# Brain Sparing Effect on Neurodevelopment in Children with Intrauterine Growth Restriction: A Systematic Review

**DOI:** 10.3390/children8090745

**Published:** 2021-08-28

**Authors:** María José Benítez-Marín, Jesús Marín-Clavijo, Juan Antonio Blanco-Elena, Jesús Jiménez-López, Ernesto González-Mesa

**Affiliations:** 1Obstetrics and Gynecology Section at Surgical Specialties, Biochemistry and Immunology Department, Medicine School, Málaga University, 29071 Málaga, Spain; egonzalezmesa@uma.es; 2Obstetrics and Gynecology Service, Virgen de la Victoria University Hospital, 29010 Málaga, Spain; 3Arts and Architecture Department, Málaga University, 29071 Málaga, Spain; jmarin@uma.es; 4General Surgery Service, Infanta Margarita Hospital, 14940 Cabra, Spain; jblanco81@me.com; 5Obstetrics and Gynecology Service, Regional University Hospital of Málaga, 29011 Málaga, Spain; jjimenezme35426@gmail.com; 6IBIMA Research Group in Maternal-Fetal Medicine, Epigenetics, Women’s Diseases and Reproductive Health, 29071 Málaga, Spain

**Keywords:** brain sparing, intrauterine growth restriction, neurodevelopment, cognitive, behavior, motor, executive functions

## Abstract

Background: Fetal growth restriction (FGR) is a pregnancy complication. Multiple studies have connected FGR to poor cognitive development, behavior disorders, and academic difficulties during childhood. Brain sparing has traditionally been defined as an adaptive phenomenon in which the brain obtains the blood flow that it needs. However, this adaptive phenomenon might not have a complete protective effect. This publication aims to systematically review the consequences of brain redistribution on neurodevelopment in children who presented with placental intrauterine growth restriction. Methods: We performed a systematic review according to PRISMA guidelines. It included studies on intrauterine growth restriction or small-for-gestational-age (SGA) fetuses, which middle cerebral artery was measured, and neurodevelopment assessed during childhood. PUBMED and EMBASE databases were searched for relevant published studies. Results: Of the 526 studies reviewed, only 12 were included. Brain sparing was associated with poor cognitive function and lower scores in IQ. Cerebral redistribution was related to better executive function and better behavior at 4 years old but not at 12 years old. Conclusions: We can assume that fetal brain sparing could not be a fully protective phenomenon. We could not find clinical differences in behavioral and executive functions because the results were heterogeneous. Some cognitive abilities could be affected in FGR brain sparing fetuses.

## 1. Introduction

Fetal growth restriction (FGR) is a pregnancy complication that occurs in approximately 10% of pregnancies worldwide. Its multiple causes can be divided into two large groups: restrictions due to placental insufficiency, or restrictions in which the placenta does not play a role in its pathophysiology (chromosomal or genetic abnormalities, congenital infections, or metabolic disorders).

Prematurity and fetal deterioration during FGR affect short-term outcomes. The TRUFFLE study described perinatal mortality and morbidity in a cohort of fetuses with FGR. Neonatal morbidity was strongly associated with gestational age at the diagnosis of FGR, with neonatal sepsis and bronchopulmonary dysplasia being more frequent. However, among preterm fetuses, no increase in necrotizing enterocolitis, germinal matrix hemorrhage, or periventricular leukomalacia was found. Adverse outcomes were more frequent at a lower gestational age or associated with hypertensive states [1].

The effects of FGR impact not only the neonatal period but also childhood and adulthood. The fetal period and childhood are sensitive stages where genomic interactions with the environment occur as organs and systems program their long-term functions. In this way, fetal programming can be altered both by prematurity or by growth restriction in hostile environments, thus increasing the risk of chronic diseases in the adult period [2]. Motor, cognitive, and behavioral development could be affected by this hostile environment.

Multiple studies have connected FGR to poor cognitive development, behavior disorders, and academic difficulties during childhood. Children with early-onset FGR tend to have a high incidence of prematurity, a risk factor for adverse neurological outcomes. Premature infants with FGR have a higher incidence of neurodevelopmental disorders compared with premature infants with adequate growth [3,4,5,6,7]. Late-onset FGR could also be related to alterations in cognitive development, academic results, and learning. Although published data are contradictory, the difference between groups tends to be minimal [8,9,10,11,12].

The relationship between Doppler abnormalities and perinatal results is well-established. Although neurological and long-term outcomes are evaluated, the results are not insightful. Brain sparing has traditionally been defined as an adaptive phenomenon in which the brain obtains the necessary resources for its proper development and functioning. Oxygen and nutrient restriction cause the brain not to acquire sufficient substrates to function. Because of this, the vascular system vasodilates its arteries to increase the blood supply. This allows the brain to obtain the supplements that it needs. However, this adaptive phenomenon might not have a fully protective effect. The presence of brain sparing in stunted children could be related to smaller head circumferences [13] and smaller brain volumes [14] at birth. This could have consequences on neurocognitive development at later ages.

Studies in animal models of FGR and humans have demonstrated changes in brain structure. Animal studies have shown a decrease in the number of neurons [15] and a modification in dendritic arbors morphology [16]. Other studies have demonstrated a delay in white matter maturation [17] and a volume decrease in white matter [18] that could affect cognitive function. Similarly, human studies have reported changes in grey matter [19,20,21], white matter [22,23], and gyrification patterns [24].

This publication aims to systematically review the consequences of brain redistribution on neurodevelopment in children who presented with placental intrauterine growth restriction. We developed a systematic review based on PRISMA statements. We describe below the results and summaries of the studies analyzed, as well as the discussion of the outcomes of our review and the subsequent conclusions.

## 2. Materials and Methods

### 2.1. Literature Search

We performed a systematic review according to Preferred Reporting Items for Systematic Review and Meta-Analyses (PRISMA) guidelines [25]. We searched PubMed and EMBASE databases to find published studies about neurodevelopment during childhood in infants with a redistribution of cardiac output to the fetal brain in the prenatal period. We carried out this search in June 2021, with no date restrictions in the publication’s reports. We pre-registered the study protocol in PROSPERO with provisional ID 272659.

For this search, the following search terms were used “fetal growth restriction”, “fetal growth retardation”, “intrauterine growth restriction”, and “small-for-gestational-age” to define growth defects; “brain sparing”, “middle cerebral artery”, and “cerebral redistribution” to define cerebral redistribution; and “development”, “cognitive aspects”, “outcomes” to defined neurodevelopment.

Our search strategy for PUBMED using a Boolean sequence was: (((Fetal growth restriction) OR (fetal growth retardation) OR (intrauterine growth restriction) OR (small for gestational age)) AND ((arteries, middle cerebral) OR (Brain sparing)) AND ((Outcomes) OR (cognitive aspects) OR (Development))). Our search strategy for EMBASE was: ((fetal) AND (‘growth’/exp OR growth) AND restriction OR (fetal AND (‘growth’/exp OR growth) AND (‘retardation’/exp OR retardation)) OR (intrauterine AND (‘growth’/exp OR growth) AND restriction) OR (small AND for AND gestational AND (‘age’/exp OR age))) AND ((‘arteries’/exp OR arteries,) AND middle AND cerebral OR ((‘brain’/exp OR brain) AND sparing) OR (cerebral AND redistribution)) AND (‘outcomes’/exp OR outcomes OR (cognitive AND aspects) OR ‘development’/exp OR development) AND (article]/lim AND (humans)/lim AND (English)/lim AND (abstracts)/lim).

### 2.2. Inclusion Criteria

The search covered studies that met the following criteria: studies on intrauterine growth restriction or small-for-gestational-age (SGA) fetuses, whenever the middle cerebral artery or its ratios were measured; studies where neurodevelopment was assessed during childhood using a well-established neurodevelopmental test (with reliability and validity) or parental reports (as long as they were standardized or validated); studies with a defined control group; studies where structural and chromosomal abnormalities were excluded; and studies in English.

Animal studies, editorials, congress and conference papers, reviews and systematic reviews, case report papers, and studies incomplete or with unpublished data were excluded.

### 2.3. Selection Process

Studies referring to neurodevelopment and cerebral redistribution were identified. Furthermore, Rayyan software was used to remove duplicates. Figure 1 shows the flow diagram for the selection process. This search initially produced 221 records by searching EMBASE, and 295 records through PUBMED. After reviewing the reference list for the more relevant studies, we included ten studies from manual searches. All 526 titles were read by two independent reviewers. Initially, 99 duplicated studies were removed, and 385 papers were excluded because the abstracts were not relevant. Forty-three studies were selected for full-text review. After fulfilling all the inclusion criteria, only 12 studies were included in the qualitative synthesis.

### 2.4. Data Collection Process

We used a manual data collection sheet to obtain and confirm data from the studies. Finally, we performed a summary table containing the main data and results. First, we defined the study group (FGR or SGA) and the criteria of brain sparing. We additionally defined the control group and its characteristics.

Secondly, we sought the neurodevelopmental tests applied and the age of the evaluation. Finally, we described the neurodevelopmental outcomes: intelligence quotient (IQ), cognitive outcomes, or behavioral outcomes.

### 2.5. Quality Assessment

The Quality Assessment Tool for Quantitative Studies was used to assess the quality of scientific evidence. The Effective Public Health Practice Project (EPHPP) developed this tool in 1998 [26]. The EPHPP assessment tool evaluates eight groups of items: (a) selection bias, (b) study design, (c) confounders, (d) blinding, (e) data collection methods, (f) withdrawals and dropouts, (g) intervention integrity, and (h) analysis. According to the score, we could classify each section and study as strong, moderate, or weak. Table 1 summarizes the EPHPP-related data for each study.

## 3. Results

### 3.1. Overall Search Results

Finally, we reviewed the 12 studies that met the inclusion criteria. All of these were longitudinal studies: 11 prospective [27,28,29,30,31,32,33,34,36,37,38] and only one retrospective [35]. Two of the studies were secondary analyses of randomized clinical trials. The first selected its participants from the TRUFFLE study in which pregnant women delivered according to antenatal monitoring strategies (reduced computerized cardiotocographic heart rate short term variation, early or late ductus venosus changes) [28,39]. The second study selected its participants (children with fetal growth restriction) from a trial in which women with severe preeclampsia, HELLP syndrome, or pregnancy-induced hypertension were randomized to receive or not receive a plasma volume expansion [29,40]. Two studies were part of multidisciplinary projects: the PIPARI study, which evaluated the development in extremely low birth weight infants [38] and the PORTO study, which analyzed the association between sonographic findings and perinatal outcomes in fetal growth restriction [27,41].

Exclusion criteria differed between studies, but all considered the presence of structural or chromosomal abnormalities as an exclusion criterion.

### 3.2. Descriptions of the Tools Used by the Studies

Brain sparing could be defined by the raised pulsatility of the middle cerebral artery or its different ratios: the cerebro–placentary ratio, which was pathological for scores below 1 or the 5th percentile, and the umbilical–cerebral ratio, which was pathological for values above 0.72. Values below the 5th percentile for the middle cerebral artery were considered pathological.

Specific neurodevelopmental tests were used to evaluate motor, cognitive, and academic outcomes at different ages. Teachers and parents reported behavior and academic achievement by standardized tests, like the Child Behavior Checklist or Behavior Rating Inventory. The Home Observation for Measurement of the Environment (HOME) method was used in one study [35] to assess the quality of stimulation and support at home by means of an interview.

The main tests used in the studies were:-Neonatal Behavioral Assessment Scale (NBAS) [30,36]. This test analyzes how newborns control their states and how the transition from one to another progresses. This test evaluates newborns’ habituation to the environment, motor system, social interaction (visual and auditory), organization and regulation state, autonomic system, and attention capacity. The scale evaluates how infants manage these vital tasks that are important for growth and development.-Bayley Scales of Infants and Toddlers [27,28,38]. This test evaluates cognition, language, and motor development. One study [38] used the second edition, which evaluates the Mental Developmental Index (MDI). The MDI estimates cognitive skills such as memory, problem-solving, vocalization, language, and social skills.-Ages and Stages Questionnaire [27,37]. This test analyzes communication, gross and fine motor skills, problem-solving and personal–social interaction.-Weschler Intelligence Scale [29,31,35]. This test assesses intelligence, verbal comprehension, working memory, processing speed, and the perceptual reasoning index. In the preschool and primary version, the test also analyzes verbal and performance IQ subscales.-Revision of the Amsterdam Children’s Intelligence (RAKIT) Test [33]. This test assesses the intelligence quotient. The disk test is a subtest that evaluates the integration of visual recognition and fine motor coordination.-Behavior Rating Inventory of Executive Function [29,31]. This test evaluates inhibitory self-control, flexibility, emergent metacognition, and the total executive function score.-Child Behavior Checklist [29,31,34]: This test evaluates the internalizing behavior (emotional reactivity, anxiety, depression, somatic complaints, and withdrawals) and externalizing behavior (attention problems and aggressive behavior).

Neurodevelopmental variables differed between studies because they were assessed at different ages and by non-identical tests. The studies try to assess cognitive, motor, and behavioral development. Table 2 summarizes the main tests used in the studies.

### 3.3. Summary of the Findings of the Studies

Almost all the studies assessed neurodevelopment during childhood. Only two studies evaluated the infant at 40 weeks corrected age by the Neonatal Behavioral Assessment Scale (NBAS) [30,36]. FGR with an abnormal middle cerebral artery presented a lower score in habituation, motor, social–interactive, and attention areas [30]. However, SGA with an abnormal middle cerebral artery only presented lower scores in motor-related variables [36]. When middle cerebral artery parameters were within the normal range, development scores were similar to those in fetuses with the appropriate weight for gestational age.

Another test was performed in children aged 2–8 years to assess cognitive neurodevelopment. The most common were the Bayley Scales of Infants and Toddlers and Ages and Stages Questionnaire [27,28,31,37,38]. These studies show that brain sparing was associated with poor cognitive development in several areas [27,28,37,38]. There was no evidence of poor motor development in fetuses without cerebral redistribution. Only one study showed worse fine and gross motor development in non-redistributed fetuses [28].

Three studies used the Weschler Intelligence Scale [29,31,35]. Only one study found lower scores in IQ when cerebral redistribution was present [35]. Scherjon et al. [33] also found a high proportion of children with IQ below 85 points in the brain sparing group at 5 years old. Lower birth weight and lower socioeconomic status were associated with a lower score in the IQ in one study [29].

Executive function was evaluated by the Behavior Rating Inventory of Executive Function in only two studies [29,31]. Fetal brain sparing was related to better executive function at 4 years old but not at 12 years old.

Behavior was mainly evaluated at an advanced age in childhood [29,34]. Only one study assessed behavior problems at a young age [31]. A behavioral assessment was conducted based on reports made by caregivers and teachers. The Child Behavior Checklist was the most frequently used test. Brain sparing was related to better total behavior and better externalizing behavior at 4 years old [31] but not at 11–12 years old [29,34]. Nevertheless, a higher incidence of behavioral problems was found in the FGR cohort compared with the general population [34]. Even school performance was not different at a late age [29,34].

Table 2 summarizes the main results of the studies.

**Table 2 children-08-00745-t002:** Main characteristics of the studies.

Study	*N*	Study Group	Control Group	Definition of Brain Sparing	Exclusion Criteria	Age Assessment	Neurodevelpment Assessment	Results
Monteith et al. (2019) [27] Ireland Longitudinal multicentre prospective cohort Secondary analysis from PORTO study	378	- FGR with normal CPR *N* = 136 - FGR with abnormal CPR *N* = 41 GA between 24 + 0 and 36 + 6	SGA *N* = 201	CPR < 1	- Birthweight < 500 g - major structural and/or chromosomal abnormalities	3-year-old	Ages and Stages Questionnaire (ASQ) Bayley Scales of Infant and Toddler Development (3rd edition)	- FGR with abnormal CPR value had significantly lower mean scores in ASQ scales and Bayley scales compared with SGA (*p* < 0.05) - FGR with normal CPR also had lower mean scores compared with SGA, but only significantly in gross and fine motor development (*p* < 0.05) - When comparing both groups of FGR, only motor score in Bayley Scales reached significance (*p* = 0.002)
Stampalija et al. (2017) [28] United Kingdom Longitudinal multicentre prospective cohort Secondary analysis from TRUFFLE study	342	Abnormal neurodevelopment outcome in FGR *N* = 310 GA between 26 + 0 and 31 + 6	Normal neurodevelopment outcome in FGR. *N* = 32		- delivery planned - major structural abnormality - fetal karyotype abnormality - <18 year-old	2-year-old	Bayley Scales of Infant and Toddler Development (3rd edition) Gross Motor Function Classification System (GMFCS) Neurodevelopmental impairment was defined as:Bayley score < 85 or cognitive delay >3 months Cerebral Palsy (GMFCS > 1) Hearing loss (hearing aids) Severe visual loss	- MCA PI and UCR Z score at study inclusion were associated with 2-year infant survival without neurodevelopmental impairment (*p* < 0.05) - CPR Z score at study inclusion, MCA I, UCR, and CPR Z score before birth, and the change of these parameters with the time were not associated with 2-year outcome - Gestational age and birth weight at delivery remained the most important factor in determining 2-year infant outcome without neurodevelopmental impairment (*p* < 0.05)
Beukers et al. (2017) [29] The NetherlandsLongitudinal prospective cohort	128	FGR (NO Doppler criteria in FGR definition) *N* = 96 GA between 24 + 0 and 34 + 6 at admission	Children with gestational age ≥ 37 weeks and birth weight ≥ 2500 g at delivery. *N* = 32	UCR > 0.72	- Several fetal distress. - Lethal fetal congenital abnormalities.	12-year-old	Weschler Intelligence Scale. Amsterdam Neuropsychological Task: Visual memory working, set shifting and focusing attention Tower London Test: Planning. Behavior Rating Inventory of Executive Function (parent report). Strengths and Weaknesses of Attention Deficit Hyperactivity Disorder Symptoms and Normal Behavior Scale (parent report) Child Behavior Checklist (parent report)	- 96% of cases had raised UCR, indicating brain sparing - Mean IQ was similar for FGR and control group (101.1 ± 16.7 vs. 105.9 ± 10.0 *p* = 0.12) - Parents of FGR reported significantly more social problems (*p* < 0.001) and FGR tend to have more attention problems (*p* = 0.07) - All executive functions, attention test performances, and parents’ reports did not differ between groups - For attention problems scores there were no significant difference between groups - UCR was not associated with any of the outcome variables - BWR and low SES were both associated with lower IQ
Figueras et al. (2011) [30] Spain Longitudinal prospective cohort	126	FGR - Normal MCA PI *N* = 29 - Abnormal MCA PI *N* = 33 Gestational age at delivery < 34 weeks *N* = 62	Singleton AGA *N* = 64	MCA PI < 5th percentile	- congenital malformations - congenital infection - chromosomal abnormalities - placental histological chorioamnionitis - infant death before 40 weeks - neurological complication	40 weeks corrected age	Neonatal Behavioral Assessment Scale (NBAS)	- Neurobehavioral score did not differ between FGR with normal MCA and the control group - Scores were significantly lower in FGR and abnormal MCA, specifically in habituation, motor, social-interactive, and attention areas (*p* < 0.05)
Richter et al. (2020) [31] The NetherlandsLongitudinal prospective cohort	25	FGR (FAC or EFW < 10th percentile or decreased fetal growth more than 30 percentiles) with FBS *N* = 11	FGR (FAC or EFW < 10th percentile or decreased fetal growth more than 30 percentiles) without FBS *N* = 14	CPR < 1	- structural or chromosomal abnormalities - multiple pregnancy - intrauterine infection	4-year-old	Weschler Preschool and Primary Scale Child Behavior Checklist (parent report).Behavior Rating Inventory of Executive Function Preschool Version (parent report)	- FBS was not associated with IQ - FBS was significantly related with better total behavior and better externalizing behavior (*p* < 0.05) - FBS tended to have better inhibitory self-control (*p* < 0.1) - Adjusted for gestational age, which is positively correlated with T-score for total behavior, total executive function, and Emergent Metacognition Index (*p* < 0.05)
Scherjon et al. (1998) [32] The Netherlands Longitudinal prospective cohort	96	Fetuses with UCR raised *N* = 34 Gestational age between 26 and 33 weeks at delivery	Fetuses with normal UCR *N* = 62	UCR > 0.72	- structural or chromosomal abnormalities	3-year-old	Ultrasound findings: intraventricular bleeding or echo densities Hempel neurodevelopmental outcome: motor system, hearing, vision, and eye movements Behavioral Aspects (parent report)	- Lower head circumference was found in infants with raised UCR (*p* < 0.02) - All infants with abnormal neurological outcomes and all but one middle neurological outcome were found in the normal UCR group (*p* = 0.23) - Gestational age was lower in abnormal neurological outcomes (*p* = 0.01) - In the normal UCR group, the association with ultrasound findings and Hempel outcomes was highly significant. No association in raised UCR group (*p* < 0.0001) - No significant differences in behavioral or language development between groups
Scherjon et al. (2000) [33] The Netherlands Longitudinal prospective cohort	73	Fetuses with UCR raised *N* = 28 Gestational age between 26 and 33 weeks at delivery	Fetuses with normal UCR *N* = 45	UCR > 0.72	- structural or chromosomal abnormalities	5-year-old	Visual Evoked Potentials (VEP) RAKIT Test: intelligent quotient Disk Test: integration of visual recognition and fine motor coordination	- Infants with normal UCR were shortening VEP latencies between 6 to 12 months (decreased 20%) (*p* = 0.0001). UCR raised group had short VEP latencies at 6 months but remained unchanged at 12 months (decreased 5%) (*p* = 0.10) - Infants with raised UCR showed a 9-point lower IQ at 5 years compared with normal group (*p* < 0.02) - 54% of infants with raised UCR were IQ < 85 compared with 20% in normal group (*p* = 0.003) - There was a positive statistical association between a greater difference in VEP latencies at 6–12 months and higher IQ
Van den Broek et al. (2010) [34] The Netherlands Longitudinal prospective cohort	89	Fetuses with UCR raised *N* = 31 Gestational age between 26 and 33 weeks at delivery	Fetuses with normal UCR *N* = 58	UCR > 0.72	- structural or chromosomal abnormalities	11-year-old	Child Behavior Checklist (parent report) Teacher’s Report Form (teacher report): based in Child Behavior Checklist	- No significant differences in the incidence of behavioral problems between groups - They found a higher incidence of behavioral problems in the cohort compared with general population - No significant difference in not adequate school performance between groups - Birth weight was more important to predictive behavioral problems (*p* = 0.003)
Bellido-Gonzalez et al. (2017) [35] Spain Longitudinal retrospective cohort	120	FGR (birth weight < 10th percentile and abnormal MCA PI < 5th percentile): FGR-A: abnormal CPR (<5th percentile) and abnormal UA PI (>95th percentile) *N* = 32 FGR-B: normal CPR and UA *N* = 27 Gestational age > 37 weeks	Term AGA	MCA PI < 5th percentile	- parental drugs consumption - multiple gestation - congenital malformation - chromosomopaties - low Apgar score - vision/hearing impairment - cerebral palsy - non-native speaker of Spanish	6–8-year-old	Wechsler Intelligence Scale for Children IV: III Woodcock–Muñoz Battery: Reading, Written Language, Mathematics Home Observation for Measurement of the Environment (HOME) methods: interview to measure the quality of stimulation and support	WISC-IV: FGR-A presented lower scores than AGA children for all measurements (*p* < 0.05) Larger differences were observed in working memory FGR-B presented lower scores than AGA only for verbal comprehension and working memory (*p* < 0.05) Academic achievement: FGR-A presented lower scores than AGA children in reading, written language, and mathematics (*p* < 0.05) FGR-B presented lower scores than AGA children only in mathematics (*p* < 0.05)
Cruz-Martinez et al. (2009) [36] Spain Longitudinal prospective cohort	120	SGA *N* = 60 Gestational age > 37 weeks	Term AGA *N* = 60	MCA PI < 5th percentile Or FMBV > 95th percentile	- Congenital malformations or chromosomopaties - *UA PI*>95th percentile	40 weeks corrected age	Neonatal Behavioral Assessment Scale (NBAS)	- SGA showed higher mean frontal FMBV values than AGA. The proportion of FMBV > 95th percentile was 35% in SGA and 5% in AGA (*p* < 0.001) - The proportion of MCA PI < 5th percentile was 15% in SGA and 1.7% in AGA (*p* < 0.01) - All neurobehavioral areas had lower scores in SGA group (*p* < 0.05) - SGA with abnormal FMBV showed lower scores in social-interactive, attention, and organization states. SGA with normal FMBV showed similar scores to AGA (*p* < 0.05) SGA with abnormal MCA showed lower scores in motor area (*p* < 0.05)
Eixarch et al. (2008) [37] Spain Longitudinal prospective cohort	222	SGA Normal MCA PI *N* = 100 Abnormal MCA PI *N* = 25 Gestational age > 37 weeks	Term AGA *N* = 97	MCA PI < 5th percentile	- Congenital malformations or chromosomopaties - UA PI > 95th percentile	24 months corrected age	Age and Stage Questionnaire (ASQ) (parent report)	- 24.7% of control group showed abnormal ASQ scores in more than one area, compared with 31% in the non-redistributed SGA and 52% in redistributed SGA groups - Differences between AGA and SGA non-redistributed group was non-significant - Differences between AGA and SGA redistributed group was significant. They showed lower scores in communication and personal-social areas (*p* < 0.05) - Compared to both SGA groups, redistributed SGA had a lower score in communication and problem-solving (*p* < 0.05)
Leppäpen et al. (2010) [38] Finland Longitudinal multicentre prospective cohort Secondary analysis from PIPARI study	83	Preterm delivery < 32 weeks or estimated birth weight < 1500 g In the secondary analysis the antenatal Doppler flow and the relationship with neurodevelopment was studied It was compared: Infants with abnormal UCR *N* = 16 Infants with normal UCR *N* = 54		UCR > 95th percentile	- congenital anomalies or a diagnosed syndrome - non-native speaker of Finnish and ⁄ or Swedish	2-year-old	Bayley Scales of Infant Development II Hammersmith Infant Neurological Examination (HINE): suboptimal < 74 Cranial nerve function, posture, movements, tone and reflexes, motor functions, behavior	- Abnormal UAPI, UCR, increased Dao PI and DAo/MCA ratio were associated with adverse cognitive performance. When the effect of confounding factor was controlled, only DAo and UCR remained statistically significant (*p* < 0.05) - When infants with normal and abnormal UCR were compared, no differences in HINE scores were found. The infants with abnormal UCR showed a lower score in MDI compared with normal UCR infants

Table 2: Summarizes the main results of the studies included in the systematic review. FGR, Intrauterine Growth Restriction (birthweight < 10th percentile and umbilical artery abnormal Doppler); SGA, Small-for-Gestational-Age (birthweight < 10th percentile and normal umbilical artery Doppler); AGA, Appropriate-for-Gestational-Age (birth weight ≥ 10th percentile); CPR, Cerebro–Placental Ratio; UCR, Umbilico–Cerebral Ratio; MCA, Middle Cerebral Artery; FMBV, Fractional Moving Blood Volume; UA, Umbilical Artery; DAo, Descending Aorta; PI, Pulsatility Index; FBS, Fetal Brain Sparing; GA, Gestational Age; BWR, Birthweight Ratio; IQ, Intelligent Quotient; SES, Social Economic Status.

### 3.4. Quality Assessment Results

According to EPHPP [26], the selection bias was moderate in all the included studies because the population could be non-representative, or the percentage of selected individuals who agreed to participate was not disclosed. Some studies reported substantial rates of withdrawal or dropout [29,31,33,37] or failed to report these data [27]. Four studies did not communicate control of the relevant confounder [31,32,33,34]. Data collection methods were reliable and valid in all studies.

## 4. Discussion

This systematic review identified 12 studies that evaluated the association between neurodevelopmental outcomes and fetal brain sparing. These studies assessed neurodevelopment at different ages and tests. Not all the studies analyzed cognitive, motor, and behavior development at the same time.

The studies analyzed reported conflicting results globally. They could detect better executive function and behavior at 4 years old, but not at 12 years old. On the other hand, non-consistent findings were detected in IQ results. Multiple studies have implicated FGR with changes in brain morphology and sizes [20,42,43] and decreases in gray matter in the cortical [20], subcortical [21], and hippocampal locations [19]. Different studies demonstrated that the gyrification pattern is different in FGR brains [24,44]. In the same way, the white matter could be decreased in some zones and increased in other zones, presenting an aberrant pattern [22,23]. Some studies have reported a small corpus callosum in the posterior region in FGR infants [45,46,47]. All these changes could be related to cognitive differences.

Traditionally, brain sparing was considered a protector phenomenon. However, multiple studies have demonstrated that this affirmation is not entirely correct. Some studies have shown differences in IQ results. Beukers et al. [29] and Richter et al. [31] did not find a substantial significance in IQ values. Another study found a worse IQ in FGR children with an abnormal cerebro–placental ratio, detecting worse verbal comprehension, perceptual reasoning, working memory, and processing speed at 6–8 years old [35]. This cohort was, however, retrospective. Scherjon et al. [33] also found a high proportion of children with IQ below 85 points in the group with raised utero–cerebral ratio at 5 years old. Perhaps these results concerning IQ are due to the assessment being made at different ages. Additionally, differences in cognitive areas (memory, communication, problem-solving, and social skills) were noted. We could detect this result in both FGR and SGA fetuses [27,37,38].

Non-behavioral differences were detected in brain sparing children at a late age. Parents reported more attention and social problems in FGR children that could not be associated with brain sparing [29,34]. When compared with the general population, a higher incidence of behavioral problems in the FGR cohort at 11 years old was found [34]. However, brain sparing was related to better behavior at 4 years old [31]. Similarly, when we analyzed the executive function, the results did not differ in the FGR cohort at a late age [29], as was the case at an early age [31]. Two reasons could be given to support this statement. Richter et al. evaluated a small number of children, which could overestimate the results [31]. As the child grows, behavioral problems could start to appear due to personality and identity development, as well to the improvement of the executive function. More studies are necessary to clarify this issue.

Of especial interest is the “Scherjon study group” [32,33,34]. These authors evaluated the same cohort for 11 years. Initially, they could not find differences in behavior or language development at 3 years old [32]. Nevertheless, they found that 54% of infants with brain sparing had an IQ below 85 points compared to 20% of children in the control group when they evaluated the same cohort at 5 years old. They also found that the visual evoked potentials remained unchanged from 6 to 12 months in the brain sparing group, although values were higher at 6 months. There was a positive association between a higher IQ and a greater difference in visual evoked potentials from 6 to 12 months [33]. However, they did not find behavioral differences at 11 years old between groups [34].

We could not identify motor problems in our review. This issue could be due to the impact of prematurity on psychomotor development, especially when the delivery takes place before 28–29 weeks [6,48]. At the same time, minor gestational age is associated with higher possibilities of intracranial hemorrhage and interventricular leukomalacia. These processes could contribute to adverse neurological outcomes [13,33]. The cerebral palsy risk ranges from 4% to 18% before 32 weeks of pregnancy, increasing as gestational age decreases [6,49].

One of the most crucial factors related to neurodevelopment were gestational age at delivery and birth weight [28,29,32,34].

Our review has some limitations. The first is the heterogeneity of the studies. The great variety in both specific tests and ages of assessment made the comparison between studies very difficult. There is a wide range of neurodevelopmental outcomes, so it is virtually impossible to compare behavior with cognitive development like memory and language skills.

The second limitation is the definition of fetal growth restriction as well as fetal brain sparing. Some studies define FGR with only measurement criteria and without Doppler criteria. There are also different definitions of fetal brain sparing. This fact makes these studies heterogeneous. The results could be underestimated as FGR as SGA fetuses are classified in the same group. We wanted to show two studies about SGA fetuses because they demonstrated that SGA fetuses with brain sparing had lower scores in specific tests. Although they tested children very early (at 40 weeks corrected age) in one of these studies, we can conclude that SGA fetuses have lower scores.

Fetal brain sparing is classically defined by middle cerebral artery measurements and ratios. The protection of the brain is a hierarchical priority. When the middle cerebral artery starts to vasodilate, the anterior cerebral artery is already affected [50,51,52]. The anterior cerebral artery supplies the anterior region of the brain, the frontal lobes. This region is crucial to cognitive functions [53]. Cruz-Martinez et al. [36] could correlate poor cognitive outcomes at 40 weeks corrected age in fetuses when they showed higher blood perfusion in the anterior region. Therefore, it is possible that some of the infants without fetal brain sparing in the control group already had a vasodilated anterior cerebral artery. This fact could confuse attempts to interpret the results.

The third limitation of the study is the lack of control of the confounders. Not all the studies adjust their results for confounding, and when they do, the control is not always the same. The environment during childhood is crucial for neurodevelopment. If a child develops under poor conditions with poor support, it is possible they will not reach their maximum potential. The fourth limitation is the lack of comparison with full-term appropriate-for-gestational-age. This could underestimate the results.

## 5. Conclusions

Based on the results of this review, we can assume that fetal brain sparing could not be a fully protective phenomenon. In severe cases, the deleterious consequences of this event on neurodevelopment could pass over the benefits of the sparing, leading to a wide spectrum of clinical manifestations. This vascular process could be useful as a marker to identify children with more risk of poorer cognitive development. We could not find clinical differences in the behavioral and executive functions between the groups. However, some cognitive abilities could be affected in FGR brain sparing fetuses. The childhood environment is vital to proper neurodevelopment, mainly in the first years of life. The detection of high-risk infants is crucial in taking preventative steps to improve neurodevelopment.

Our review has some weaknesses, such as the heterogeneity of the studies, the heterogeneity in the definition of fetal growth restriction or fetal brain redistribution, the lack of control of the confounders, and the selection of only studies published in English as well as only studies with published data. More homogeneous studies are necessary to investigate the role of brain sparing in predicting poor neurodevelopment.

## Figures and Tables

**Figure 1 children-08-00745-f001:**
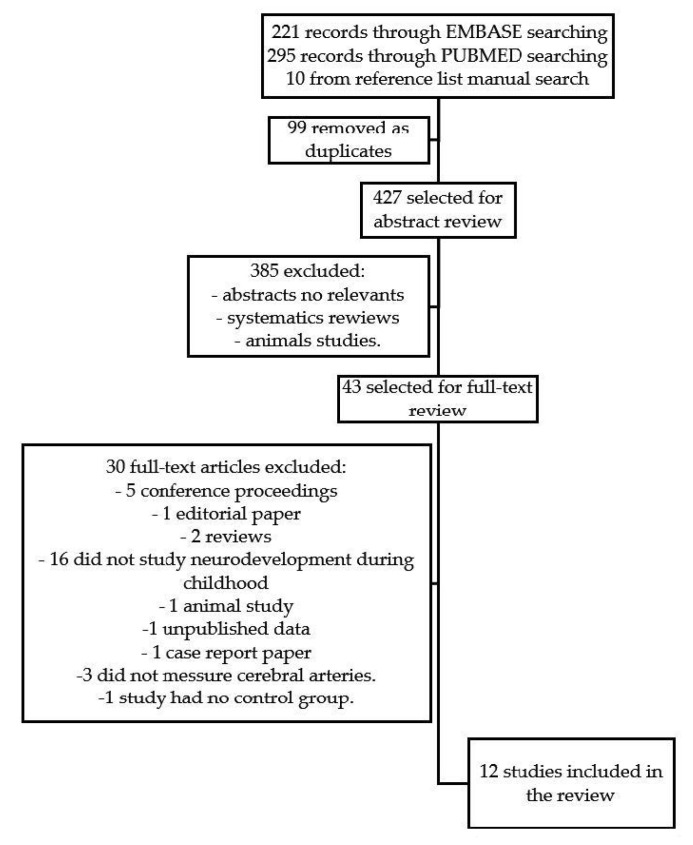
Flow diagram for the review.

**Table 1 children-08-00745-t001:** EPHPP rating for each study.

Study	Selection Bias	Study Design	Confounders	Blinding	Data Collection Method	Withdrawals and Dropouts	Global Rating
Monteith et al. (2019) [27]	Moderate	Moderate	Strong	Moderate	Strong	Weak	Moderate
Stampalija et al. (2017) [28]	Moderate	Moderate	Moderate	Moderate	Strong	Moderate	Strong
Beukers et al. (2017) [29]	Moderate	Moderate	Strong	Moderate	Strong	Weak	Moderate
Figueras et al. (2011) [30]	Moderate	Moderate	Moderate	Moderate	Strong	Strong	Strong
Richter et al. (2020) [31]	Moderate	Moderate	Weak	Moderate	Strong	Weak	Weak
Scherjon et al. (1998) [32]	Moderate	Moderate	Weak	Moderate	Strong	Strong	Moderate
Scherjon et al. (2000) [33]	Moderate	Moderate	Weak	Moderate	Strong	Weak	Weak
Van den Broek et al. (2010) [34]	Moderate	Moderate	Weak	Moderate	Strong	Strong	Moderate
Bellido-Gonzalez et al. (2016) [35]	Moderate	Moderate	Strong	Moderate	Strong	Strong	Strong
Cruz-Martinez et al. (2009) [36]	Moderate	Moderate	Strong	Moderate	Strong	Strong	Strong
Eixarch et al. (2008) [37]	Moderate	Moderate	Strong	Moderate	Strong	Weak	Moderate
Leppäpen et al. (2010) [38]	Moderate	Moderate	Strong	Moderate	Strong	Strong	Strong

## Data Availability

Not applicable.

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
