# Peer review of "Brain Sparing Effect on Neurodevelopment in Children with Intrauterine Growth Restriction: A Systematic Review"

_children, 2021, doi:10.3390/children8090745_

Round 1

Reviewer 1 Report

The authors present a systematic review on whether prenatal evidence of brain sparing in the setting of fetal growth restriction or SGA is associated with differences in neurodevelopment in childhood. This is an interesting question and the authors provide a helpful summary of findings addressing this research question. Very good work!

Abstract:

  • Consider using FGR rather than IUGR. FGR is now endorsed by both ISUOG and SMFM.
  • I suggest that language on brain sparing focus on redirecting blood flow rather than the brain “obtaining nutrients that it needs”, since brain sparing is defined by differences in flow and nutrient delivery/transport is not being evaluated in this study.
  • “Brain sparing could be associated with poor…” does mean that it was associated, or not, or conflicting results? The use of “could be” here is too vague.
  • By “execution function” I think the authors mean “executive function”?
  • The sentence “behavior was evaluated in three studies;” seems incomplete. Is it in reference to the sentence before it? Or should there be something after it (it ends with a semicolon)?
  • The conclusion is confusing. Does “substantial” mean clinically or statistically significant? Were they inconsistent findings? Please try to clarify what this means.

Intro:

  • The description of the TRUFFLE study makes several comparative statements “a higher incidence of sepsis…was detected” without explaining what group TRUFFLE participants are being compared to.

Methods

  • Please include the full search strategy in a supplemental file to support reproducibility.
  • Did the authors review the reference lists of included studies for additional potentially relevant titles?
  • Consider evaluating the results reported in conference abstracts and unpublished data (if they otherwise meet inclusion criteria) together with the included results in a funnel plot to evaluate for publication bias. A small enough number of titles were excluded for being conference proceedings or unpublished (n=6) that this should be achievable.
  • Was eligibility confirmed by more than one investigator or were all titles reviewed in duplicate to minimize errors?
  • Was the study protocol pre-registered at a site such as the PROSPERO database?

Results

  • Please report the global EPHPP rating for each included study. This would be well suited to a table or figure. Also, this should be presented early in the results rather at the end so the reader knows about potentially limited study quality before reading the results.
  • The descriptions of the included studies’ findings are intermixed with descriptions of the assessment tools used in studies, making the findings hard to find and interpret. I suggest having defined sub-headings within the results that describe 1) the overall search results, 2) descriptions of the tools used by studies, and 3) summary of the findings of the studies on brain sparing and neurodevelopmental outcomes.
  • Line 184 “these studies show that brain sparing could be associated…” What does “could be associated” mean? Do the studies report conflicting or inconsistent results? Almost but not quite statistically significant? Please clarify.
  • Line 196 – I think by “execution function” the authors mean “executive function”
  • The results would be much clearer with a summary table showing whether each study found an association of brain sparing with outcomes. This would be a more concise version of table 1, and would help the reader better understand the studies together.

Discussion:

  • The discussion section needs a very brief, concise summary of the findings in the 1st “Overall, studies found that brain sparing was / was not associated with …” or “studies reported conflicting results on whether…”
  • I do not think the findings support the conclusion that brain sparing is not protective, since brain sparing may only occur in severe cases and still have a protective effect that cannot completely mitigate the severity of the clinical presentation. The study designs of the included studies do not allow inference of whether brain sparing is protective or harmful but only whether there is an association with better/worse outcomes.
  • Acknowledge weaknesses:
    • Possible bias from restricting the search to published studies and those published in English.
    • No pre-registration of study protocol
    • Lack of duplicate eligibility screening and data extraction by multiple authors
  • Is the finding that brain sparing was associated with better overall behavior and externalizing behavior conflicting with other findings showing that brain sparing is associated with worse neurodevelopmental outcomes? This deserves more discussion.

Reviewer 2 Report

Thank you for the opportunity to read the paper. The subject of the article is very interesting and important. The manuscript fits the journal and contains an interesting idea, however, there are several points that I think the authors should definitely consider.
1) The title of the article is consistent with the content 2) The introduction should clearly define the purpose of the paper and its significance. Moreover, the end of the section lacks a description of the structure of the paper, which is very useful for readers line 82
3) I recommend providing more detailed information about the methodology, such as the modeling procedure in the cited studies, etc. line tabele 1.
4) Please provide limitations and future research needs in the conclusions part.
I hope the authors can reconsider what is the contribution of the manuscript, who should use these results, and for what purpose, what are the limitations of the cited research.

Best regards
